# AbobotulinumtoxinA Doses in Upper and Lower Limb Spasticity: A Systematic Literature Review

**DOI:** 10.3390/toxins14110734

**Published:** 2022-10-26

**Authors:** Alexis Schnitzler, Clément Dince, Andreas Freitag, Ike Iheanacho, Kyle Fahrbach, Louis Lavoie, Jean-Yves Loze, Anne Forestier, David Gasq

**Affiliations:** 1PRM Department, GH St Louis Lariboisière F. Widal, Paris University, 75010 Paris, France; 2Ipsen, 92100 Boulogne-Billancourt, France; 3Evidera, London W6 8BJ, UK; 4Evidera, Waltham, MA 02451, USA; 5Evidera, Montreal, QC H4T 1V6, Canada; 6Department of Functional Physiological Explorations, University Hospital of Toulouse, 31400 Toulouse, France; 7ToNIC, Toulouse NeuroImaging Center, Université de Toulouse, Inserm, UPS, 31300 Toulouse, France

**Keywords:** botulinum toxins, muscle hypertonia, muscle spasticity, injections, intramuscular, central nervous system diseases

## Abstract

Disabling limb spasticity can result from stroke, traumatic brain injury or other disorders causing upper motor neuron lesions such as multiple sclerosis. Clinical studies have shown that abobotulinumtoxinA (AboBoNT-A) therapy reduces upper and lower limb spasticity in adults. However, physicians may administer potentially inadequate doses, given the lack of consensus on adjusting dose according to muscle volume, the wide dose ranges in the summary of product characteristics or cited in the published literature, and/or the high quantity of toxin available for injection. Against this background, a systematic literature review based on searches of MEDLINE and Embase (via Ovid SP) and three relevant conferences (2018 to 2020) was conducted in November 2020 to examine AboBoNT-A doses given to adults for upper or lower limb muscles affected by spasticity of any etiology in clinical and real-world evidence studies. From the 1781 unique records identified from the electronic databases and conference proceedings screened, 49 unique studies represented across 56 publications (53 full-text articles, 3 conference abstracts) were eligible for inclusion. Evidence from these studies suggested that AboBoNT-A dose given per muscle in clinical practice varies considerably, with only a slight trend toward a relationship between dose and muscle volume. Expert-based consensus is needed to inform recommendations for standardizing AboBoNT-A treatment initiation doses based on muscle volume.

**Plain Language Summary:** People with specific diseases or injuries of their nervous system may develop permanent stiffening of muscles in their arms and/or legs, known as spasticity; this can follow, for example, a stroke, brain damage from head injuries, or certain neurological diseases and impact mobility. Spasticity can be reduced by periodic injections of a drug called abobotulinumtoxinA (AboBoNT-A) into affected muscles; this treatment reduces muscles’ ability to contract, thereby lessening the stiffening. However, there are concerns physicians may give insufficient doses of AboBoNT-A through fears about excessive dosing; this wariness probably reflects the lack of both agreement among medical experts and clear guidance in the product literature about how to adjust doses according to the volume (i.e., bulk) of different muscles. Given this uncertainty, we carried out a systematic review to identify and analyze published information on the doses of AboBoNT-A used to inject different muscles. Specifically, we searched standard databases and websites of three scientific conferences for research on patients with spasticity (from any cause) treated with AboBoNT-A, either as part of a clinical trial or during everyday medical care. Thus, 49 relevant studies were identified for inclusion in the review. Evidence from these studies suggested that the AboBoNT-A dose given per muscle in clinical practice varies greatly, with little or no link between dose and muscle volume. Thus, there is a need for agreement between experts so that clear recommendations can then be drawn up on how best to choose the appropriate starting dose of AboBoNT-A for a particular muscle volume.

## 1. Introduction

Limb spasticity is a disabling condition characterized by muscle stiffness, pain and occasionally sudden uncontrollable movements (muscle spasms) of the upper or lower limbs [1,2]. Here, spasticity is used as a standard term to refer to the three components of muscle hypertonia: spasticity, spastic dystonia and spastic co-contractions; it develops in the lower limbs of almost half of adults who experience a stroke and can also occur following traumatic brain injury, cerebral palsy or as part of progressive diseases causing upper motor neuron lesions such as multiple sclerosis [1,2]. Clinical studies have shown that treatment with abobotulinumtoxinA (AboBoNT-A), a neurotoxin that causes muscle weakness by blocking the release of acetylcholine at the neuromuscular junction, reduces both upper and lower limb spasticity in adults, with a good tolerance profile [3,4]. However, there are concerns around AboBoNT-A treatment initiation that can prompt clinicians to be over-cautious in using the therapy, so resulting in the administration of doses that are inadequate for patients’ needs; this situation is likely due to the wide dose ranges per muscle described in the summary of product characteristics [5] or cited in the literature, the lack of consensus on adjusting these doses according to several factors (e.g., muscle volume, etiology and severity of spasticity, muscle structure), and/or the high quantity of toxin available for injection; it has been demonstrated that at maximal dose per label, higher toxin quantity (2 to 3 fold) could be injected over a single session with AboBoNT-A in adults, compared with other formulations, allowing treatment of a greater number of target muscles [6]. Current French clinical guidelines for the treatment of spasticity did not provide recommendations about AboBoNT-A dose to be injected per specific muscle [7], while clinical guidelines from the Royal College of Physicians in the United Kingdom reported muscle-specific recommendations with large dose ranges for several muscles (e.g., biceps brachii: 100–300 U) [8]. Recently, consensus guidelines for botulinum toxin therapy from the Interdisciplinary Working Group for Movement Disorders (IAB) did not consider AboBoNT-A because this drug was said to have different potency labeling compared with the other two main botulinum toxins A (onabotulinumtoxinA and incobotulinumtoxinA) [9]; this is keeping with a general acceptance that none of these toxins can be compared directly since they each contain a different quantity of neuroactive toxin and dose units are not interchangeable between them [6].

Given the uncertainties around current clinical practice, this study aimed to gather evidence on intramuscular dosages of AboBoNT-A used by healthcare professionals. Specifically, it involved conducting a systematic review to explore data from published interventional and observational studies of such treatment in adults with upper or lower limb spasticity regardless of the etiology of this condition. 

## 2. Results

### 2.1. Study Selection

The literature searches identified 1781 unique records from the electronic databases. Of these, 349 abstracts met the criteria for full-text review, which determined that 53 of the publications were eligible for inclusion in the systematic review. In addition, 3 eligible conference abstracts were identified from the grey literature searches of conference proceedings, so resulting in a total of 56 publications (see Figure 1). Most of these (49 of 56) were the primary publications for unique studies, with the rest (7 of 56) being deemed related publications because of a clear overlap with population/patient samples reported in some of the primary publications, based on details of the trial/cohort name, and enrollment years.

### 2.2. Study Characteristics

Most of the 49 primary studies included in the systematic review were conducted in Europe (n = 30), with the rest being international studies or from the Middle East/Asia (n = 7 each), Oceania (n = 3), Africa (Tunisia; n = 1), and South America (Brazil; n = 1). About half of the studies were randomized controlled trials (RCTs; n = 24), with the rest being observational real-world studies (n = 18), single-arm trials (n = 6) or non-randomized trials (n = 1). Sample sizes across studies ranged from nine to 456 patients, with most studies (36/49; 73%) enrolling fewer than 100 patients each. Most studies (31/49; 63%) reported on patients with upper limb spasticity, while 4 studies reported on patients with upper or lower limb spasticity. Overall study characteristics of the included studies are shown in Table 1.

The mean age of patients varied between 41.6 and 69 years. Information on the underlying etiology of spasticity was available for 44 studies. Most included patients with limb spasticity due to stroke or brain injury (38 studies), five studies included patients with multiple sclerosis or other disorders causing upper motor neuron lesions (e.g., degenerative myelopathy, Strümpell–Lorrain disease), and one study included a population with head or spinal cord injuries, or those who had undergone neurosurgery.

### 2.3. Risk of Bias

The 46 full-text studies included in the systematic review included 23 RCTs, seven quantitative non-randomized studies, and 16 quantitative descriptive studies (Appendix A). The risk-of-bias assessment indicated no concerns regarding study quality across the 23 RCTs, but not all assessment questions could be fully answered for the non-randomized and quantitative descriptive studies. However, these data were not considered to have a material bearing on the findings of the systematic review because the primary focus of the quality-assessment tool was the impact of study quality on treatment outcomes, rather than on assigned treatment dosing (the focus of the review).

### 2.4. Treatment Information Available from Included Studies

Studies were selected for inclusion in the systematic review on the basis that they reported a mean/median AboBoNT-A dose, a fixed dose (i.e., patients received a specific dose for a specific muscle) or a dose range for a specific muscle. Although some studies also included other treatment arms (e.g., placebo/control or another botulinum toxin A treatment), only data relating to AboBoNT-A were extracted. The data on the administration of AboBoNT-A derived from individual studies for analysis in the systematic review are presented in Appendix B.

The range of concomitant treatments used with AboBoNT-A across the studies included other medications, rehabilitation programs (e.g., physiotherapy and occupational therapy), and electrical stimulation, and one study used robot-assisted gait training to improve patient walking ability [48].

### 2.5. Dose per Muscle Volume Analysis

The 49 unique clinical trials and real-world practice studies collectively reported AboBoNT-A dose information across 50 specific muscles of both limbs. The relationship between muscle volume and AboBoNT-A dose given in these studies was explored through scatter plots. For these plots, the specific muscles injected in each study were assumed to have the average muscle volume in cm^3^ that was reported for upper-limb muscles in Holzbaur et al., 2007 [59] and lower-limb muscles in Handsfield et al., 2014 [60]. Accordingly, dose values were plotted only for those muscles for which the muscle volume was available. For example, no information on the volume of the adductor pollicis muscle was available, and therefore AboBoNT-A dose values reported for this muscle were not included in the upper-limb scatter plot. Based on muscle-volume clusters on the volume-dose plots, muscles were grouped into three volume categories (small, medium, and large). In the upper limb, large-, medium-, and small-volume muscles had a volume of ≥100 cm^3^, 20–99 cm^3^, and <20 cm^3^, respectively. In the lower limb, the respective volumes were ≥400 cm^3^, 100–399 cm^3^, and <100 cm^3^.

#### 2.5.1. Upper Limb

In the upper limb, mean, median, or fixed doses were most commonly reported for the flexor digitorum profundus (23 studies), biceps brachii (20), flexor carpi ulnaris (20), flexor digitorum superficialis (20), flexor carpi radialis (19), brachioradialis (15), and pectoralis major (14).

Wide dose ranges were found across studies, even when accounting for average muscle volume. In the small-volume muscle group, AboBoNT-A mean and median doses ranged from 47 U to 150 U, and 25 U to 200 U, respectively (Table 2). In the medium-volume group, mean and median doses ranged from 62.5 U to 200 U, and 50 U to 300 U, respectively. In the large-volume muscle group, mean and median doses ranged from 50 U to 400 U, and 75 U to 300 U, respectively.

A positive correlation between AboBoNT-A dose and average muscle volume was more clearly identified when including only studies that reported the number of patients injected with AboBoNT-A into a specific muscle (Figure 2). The mean/median dose generally ranged from 100 U to 200 U for small- and medium-volume muscles, when considering only values for 50 or more treated patients. A similar trend was observed for the large-volume muscle group, although the mean/median AboBoNT-A dose was more likely to be around 200 U to 250 U, particularly in larger muscles with an average volume of 250 cm^3^ or more. These findings should, however, be interpreted with caution as some studies reporting on upper limb muscles (6 of 34) were not included in the plot as they did not report the number of patients receiving AboBoNT-A treatment per muscle. Of note, the plots did not provide any evidence to suggest differences between interventional and RWE studies in the relationship between muscle volume and dose.

#### 2.5.2. Lower Limb

In the lower limb, mean, median, or fixed doses were most commonly reported for the tibialis posterior (10 studies), and soleus, lateral and medial gastrocnemius (8 studies each). Data for each of the remaining muscles were mostly available from one or two studies only.

In the small-volume muscle group, only two muscles were included in the scatter plot since the average volume was not available for the three other muscles reported in some studies. The mean dose for the flexor digitorum longus (average muscle volume: 30 cm^3^) and the flexor hallucis longus (average muscle volume: 78.8 cm^3^) ranged from 106 U to 233.3 U, and 94.9 U to 164 U, respectively. Relatively consistent mean-dose ranges were reported for medium-volume muscles, averaging 85 U to 372.7 U. In the large-volume muscle group, mean doses ranged from 88 U to 495.3 U (or up to 750 U if fixed doses were included).

When considering only values for groups of more than 50 patients receiving AboBoNT-A, the dose generally ranged between 100 U and 180 U for small-volume muscles, and between 100 U and 300 U for medium-volume muscles (Figure 3). Although data on larger muscles were scarce, larger studies (n > 50) tended to report a general range of 300 U to 500 U. As for the upper limb, these findings should be interpreted with caution given that some studies reporting on lower limb muscles (2 of 19) were not included on the plot as they did not report on the number of patients treated with AboBoNT-A. As for the upper limb, the plots did not provide any evidence to suggest differences between interventional and RWE studies in the relationship between muscle volume and dose.

## 3. Discussion

AboBoNT-A was approved by the United States Food and Drug Administration in 2015 for adults with upper limb spasticity, and it received label extensions for lower limb spasticity in children and adults in 2016 and 2017, respectively, and for upper limb spasticity in children in 2019. AboBoNT-A is also approved in Europe for upper and lower limb spasticity. However, the establishment of the drug as a recommended treatment option for adults with spasticity has occurred in the absence of published consensus on whether or how dosing should be adjusted in line with the volume of the target muscles, within the broad licensed dose ranges; this is the basis of concerns that treatment of such patients may be suboptimal due to the administration of inadequate doses. Although they should be the reference in terms of dosing, licensed dose ranges have been based on initial evidence from clinical trials, which helps to explain why they are wide. Against this background, the current review aimed to systematically summarize data on AboBoNT-A dose given per specific muscle of the upper and lower limb in adults with limb spasticity irrespective of underlying etiology or country in which the primary study was conducted. The results were intended to explore the extent of variability in AboBoNT-A prescribing in clinical practice, from both a clinical-trial and a real-world perspective.

Overall, there was no evidence of a strong relationship between muscle volume and AboBoNT-A dose, with wide dose ranges being reported for the same muscle or across muscles of a similar volume. For the upper limb, dose ranges were relatively consistent across small- and medium-volume muscles (mean/median 25 U to 300 U), and slightly higher doses were reported for large-volume muscles (mean/median 50 U to 400 U). Slightly higher doses and greater dose ranges were reported for the lower limb, presumably reflecting the larger volume of the muscles and the greater heterogeneity of this muscle-group category with regard to muscle volume.

This systematic review had some key strengths. To our knowledge, it is the first research to analyze potential inter-relationships between AboBoNT-A doses being used for spasticity and the volume of the injected muscles in adults, and so it targets an important gap in the literature. In a recently published systematic review and meta-analysis of clinical trials of the effects of AboBoNT-A on the Modified Ashworth Scale score in patients with stroke-related spasticity, Ojardias and colleagues reported a D_50_ of 491.7 U for large-volume muscles (arm muscles injected up to the elbow, and leg muscles down to the ankle) and 108 U for small-volume muscles (other muscles) [61]. Crucially, however, no further relationship analysis between muscle volume and the AboBoNT-A dose injected was reported. Other strengths of our review included its assessment of the evidence from both real-world and interventional studies, thereby ensuring the capture of relevant evidence on dosing practice across a broad range of practice settings and clinical scenarios. Moreover, the review included no limitations as to the etiology of spasticity or the specific muscles involved, to help ensure the representativeness and potential generalizability of its findings.

The review also had some limitations. First, there was considerable variability in how AboBoNT-A doses were reported across studies, and several studies did not report on the number of patients receiving an AboBoNT-A injection in a specific muscle. Furthermore, to increase the availability of AboBoNT-A dose data, this review included mean, median, and fixed values (i.e., where all patients received the same dose for the specific muscle). Mean values are, however, prone to outliers, which was evident in their considerably wider dose ranges compared to those for median values. Second, the variability of AboBoNT-A doses per muscle volume across the included studies could be explained by factors not captured by this systematic review (e.g., the severity of hypertonia, type of symptomatology, the dilution of the toxin prior to injection, pennate or fusiform muscle, and different study objectives).

## 4. Clinical Opinion

Despite the limitations of this systematic review, its findings indicate a pressing need for clear guidance on AboBoNT-A dosing for adults with spasticity. With this in mind and based on our practice, we propose “easy to remember “narrow AboBoNT-A dose ranges to be injected in first intention into muscles of different volume categories, as listed in Table 3. These are for first-intention AboBoNT-A treatment in botulinum toxin-naïve patients. In general, we observed that the suggested dose is 1 to 1.5 times the muscle volume (100 to 150 U for a muscle volume of 100 cm^3^) for both upper and lower limbs. These rather conservative dose ranges have a well-established safety profile since they are within French SmPC dose ranges (for in-label muscles) [5]. However, doses can be adjusted according to efficacy and the desired effect. Dose increases are possible in the absence of safety concerns and if there is an insufficient effect from a previous dose. These dose ranges are starting-points and the dose to be used may be adjusted based on the following factors:(1)etiology of the hypertonia;(2)type of hypertonia (i.e., spasticity vs. dystonia);(3)severity of hypertonia;(4)time post onset of spasticity;(5)structure of the muscle (i.e., smaller doses are needed to target the neuromuscular junctions in a long muscle such as the biceps brachii [neuromuscular junctions are all in the same place] whereas in bipennate muscles [e.g., rectus femoris, gastrocnemii] the junctions are much more disseminated such that greater doses may be required);(6)individual patient characteristics (e.g., size, weight, presence of fixed contractures, fibrosis);(7)whether the function associated with the muscle is impaired or not (e.g., iliac muscle for movement of the lower limb);(8)desired duration of action.

**Table 3 toxins-14-00734-t003:** Proposed abobotulinumtoxinA dose ranges per muscle volume ^∆^.

Range of AboBoNT-A Doses (U)	Muscle Volume (SD) * (cm^3^)	Dose Ranges According to French Label (U) [5]	Muscles (Off-Label Use in Italic)
Upper Limb			
200–300	380.5 (157.7)	NA	*Deltoideus* **
	372.1 (177.3)	150–300	Triceps brachii^†^
	290.0 (169.0)	100–300	Pectoralis major
	262.3 (147.2)	150–300	Latissimus dorsi
	164.5 (63.9)	75–300	Subscapularis
	143.7 (63.7)	50–400	Brachialis
	143.7 (68.7)	50–400	Biceps brachii
100–200	91.6 (39.3)	100–200	Flexor digitorum profundus
	74.2 (27.4)	100–200	Flexor digitorum superficialis
	65.1 (36.0	50–200	Brachioradialis
	50.0 (20.4)	NA	*Supraspinatus*
	38.4 (17.2)	45–200	Pronator teres
	37.1 (13.6)	25–200	Flexor carpi ulnaris
	34.8 (17.1)	25–200	Flexor carpi radialis
	32.7 (16.3)	NA	*Teres major*
	28.0 (13.9)	NA	*Teres minor*
	17.1 (6.3)	20–200	Flexor pollicis longus
	17.0 (7.4)	NA	*Extensor carpi ulnaris*
25–100	11.9 (5.7)	NA	*Abductor pollicis longus*
	11.2 (5.8)	NA	*Pronator quadratus*
	6.6 (3.4)	NA	*Extensor pollicis longus*
	NA	25–50	Thenar Eminence muscles ^‡,§^
	NA	NA	*Hypothenar Eminence muscles* ^‡,¥^
	NA	NA	*Dorsal and Palmar Interossei* ^‡^
Lower Limb			
200–400	849.0 (194.7)	100–400	Gluteus maximus
	830.9 (194.3)	NA	*Vastus lateralis*
	559.8 (129.4)	100–300	Adductor magnus
	438.2 (91.6)	300–550	Soleus
	274.8 (89.9)	NA	*Psoas*
	270.5 (56,6)	NA	*Vastus intermedius*
	269 (64.3)	100–400	Rectus femoris
	257.4 (61.8)	100–450	Medial gastrocnemius
	245.4 (54.2)	NA	*Semimembranosus*
	206.5 (48.4)	NA	*Biceps femoris (long head)*
150–200	186 (47.0)	NA	*Semitendinosus*
	176.8 (41.6)	NA	*Iliacus*
	163.7 (41.9)	NA	*Sartorius*
	162.1 (43.7)	50–150	Adductor longus
	150 (42.2)	100–450	Lateral gastrocnemius
	135.2 (27,5)	NA	*Tibialis anterior*
	104.8 (22.3)	100–250	Tibialis posterior
100–150	104 (24.8)	100–200	Gracilis
	104 (25.8)	50–150	Adductor brevis
	100.1 (32.0)	NA	*Biceps femoris (short head)*
	102.3 (21.6)	NA	*EDL + EHL + peroneu tertius*
	78.8 (23.1)	50–200	Flexor hallucis longus
	30 (8.2)	50–200	Flexor digitorum longus
25–100	NA	50–100, 50–200	Intrinsic muscles (*abductor hallucis*, flexor digitorum brevis, flexor hallucis brevis, *extensor digitorum brevis*) ^‡^
	NA	NA	*Interossei* ^‡^

∆ These proposals are intended to facilitate first intention AboBoNT-A treatment in botulinum toxin-naïve patients, not to be taken directly as clinical recommendations. * From Holzbaur et al., 2007 [59] for the upper limb and Handsfield et al., 2014 [60] for the lower limb. ** In practice, lower doses are injected in either anterior, medium or posterior deltoid. ^†^ In practice, lower doses are injected in either long or medial/lateral head. ^‡^ The exact volume of this muscle is unknown. Ranking is arbitrary. ^§^ Adductor pollicis, *opponens pollicis, flexor pollicis brevis, abductor pollicis brevis. ^¥^ Opponens digiti minimi, abductor digiti minimi, flexor digiti minimi brevis, palmaris brevis*. Legend: EDL, Extensor digitorum longus; EHL, Extensor hallucis longus; NA, not available; SD, standard deviation; U, unit.

## 5. Conclusions

The AboBoNT-A doses used to treat adults with upper or lower limb spasticity reported in the literature varied considerably across muscles, having only a moderate association with muscle volume. Expert-based consensus is needed to inform recommendations for standardizing initial dose ranges of AboBoNT-A treatment based on muscle volume in such patients.

## 6. Materials and Methods

This systematic review was conducted in accordance with standards of established guidelines (i.e., PRISMA) [62] and the Cochrane Handbook for Systematic Reviews of Interventions [63]).

### 6.1. Eligibility Criteria

#### 6.1.1. Types of Studies

Clinical trials and real-world evidence studies were of interest; however, articles indexed as case reports, reviews, letters, or news were excluded from the searches and during screening. 

#### 6.1.2. Types of Participants

Studies including only adults (age > 18 years) with upper or lower limb spasticity, regardless of etiology were considered eligible for this systematic review.

#### 6.1.3. Types of Interventions

Studies investigating AboBoNT-A treatment and reporting a mean/median dose of AboBoNT-A or a dose range for a specific muscle were considered. Studies that reported doses only for muscle groups, rather than for specific muscles were not eligible.

### 6.2. Information Sources

Searches were conducted in MEDLINE, MEDLINE In-Process and Embase via Ovid SP (https://ovidsp.ovid.com, accessed on 12 November 2020). The following conferences were also searched for relevant abstracts from 2018 to 2020 meetings: (1) International Society of Physical and Rehabilitation Medicine (2018: Paris, France; 2019: Kobe, Japan; 2020: virtual); (2) World Congress for Neurorehabilitation (2018: Mumbai, India; 2020: virtual); (3) Toxin’s (International Neurotoxin Association; 2019: Copenhagen, Denmark; 2021: virtual). In addition, the bibliographies of relevant systematic reviews published in the past three years and identified during the screening of material retrieved by the searches were cross-checked as a quality-assurance step to identify any relevant studies that were not identified through the electronic database searches.

### 6.3. Search Strategy

Searches were based on separate search terms for upper and lower limb spasticity and AboBoNT-A as treatment. The search strategy involved a combination of Medical Subjects Headings (extremities/arm/leg/limb/muscle spasticity/muscle hypertonia/dystonia/spasticity/stroke/cerebral palsy/cerebrovascular accident/multiple sclerosis/spinal cord injury/spinal cord injuries) and the keywords “botulinum toxin A,” “dysport,” “abobotulinumtoxinA,” “abobotulinum toxin type A,” “abobotulinum toxin A,” “botulinum a toxin,” “botulinum toxin type a,” “type a botulinum toxin$,” “clostridium botulinum toxin type a,” “clostridium botulinum a toxin botulinum neurotoxin a,” “limb or arm or leg or arms or legs or extremit$,” “spastic$ or hypertonic or hypertonia$ or dystonia$ or dystonic,” “cerebral palsy/stroke or post-stroke or spinal cord injury* or multiple sclerosis” and a combination thereof. No limitations on the publication date were applied, and the searches were limited neither by language nor geography.

### 6.4. Selection Process

Once the literature searches had been conducted and duplicate records across the databases had been removed, each title and abstract identified was screened by two independent investigators according to the inclusion/exclusion criteria. The full-text articles of studies accepted at the abstract level were retrieved for further review. The full-text screening was conducted by two independent investigators using the same inclusion and exclusion criteria that had been applied during abstract screening. Accepted articles needed to meet all of the inclusion criteria and none of the exclusion criteria. During both rounds of screening, discrepancies were resolved through discussion between investigators, and a third, senior investigator was consulted if necessary.

### 6.5. Data Collection Process

Extraction of data from the included studies was performed using a Microsoft Excel^®^-based data extraction template. The data extraction was conducted by one investigator, and reviewed by a second, senior investigator to ensure consistency and accuracy as a validation step. Any discrepancies were resolved in discussion with a third investigator by comparing the collected data with the information provided by the full paper or abstract. Extracted items included baseline characteristics (population and disease etiology), and information related to treatment with AboBoNT-A (dose and type of value [mean, median, fixed, range], upper and/or lower limb, muscle treated). Patient and treatment characteristics were only extracted for the patient group receiving AboBoNT-A; information on comparator treatments or comparative outcome data were not extracted. Data from any study that was represented in multiple articles (including interim and/or final/complete results, post-hoc or subgroup analyses) were extracted as being from a single study.

### 6.6. Study Risk-of-Bias Assessment

Quality assessment of qualitative research, RCTs, non-randomized studies, quantitative descriptive studies, and/or mixed methods studies included in this systematic review was conducted by using the Mixed Methods Appraisal Tool (MMAT) version 2018, Canadian Intellectual Property Office, Industry Canada. [64]. The MMAT can be used to appraise the quality of various types of empirical studies (i.e., primary research based on experiment, observation or simulation). A single study-design category is selected for each included study and appraised with the respective questions per category. No overall score is assigned with this tool; answers to questions relevant to each category are assigned as “yes,” “no,” or “can’t tell.” Note that, in order to operate the tool, assessment of the quality of the included studies could be conducted only for the objectives and outcomes for which the studies were designed rather than specifically for the dosing data they provided for the systematic review. Conference abstracts were not quality-assessed due to the limited information available in them.

### 6.7. Data Analysis and Synthesis

The relationship between muscle volume and AboBoNT-A dose given in the included studies was explored through scatter plots. The specific muscles injected in each study were assumed to have the average muscle volume in cm^3^, as reported in Holzbaur et al., 2007 for upper-limb muscles [59] and Handsfield et al., 2014 for lower-limb muscles [60]. Based on muscle-volume clusters on the volume-dose plots, individual muscles were grouped into three volume categories (small, medium, and large). In the upper limb, large-, medium-, and small-volume muscles had a volume of ≥100 cm^3^, 20–99 cm^3^, and <20 cm^3^, respectively. In the lower limb, the respective volumes were ≥400 cm^3^, 100–399 cm^3^, and <100 cm^3^. Across studies and for muscles for which sample size was reported, average AboBoNT-A doses (mean, median or fixed-dose values, depending on data availability) were plotted against the average muscle volume to explore interrelationships between these two variables. Dose values were plotted only for muscles for which the average muscle volume was available. The dot size on the plot was weighted by sample size for each muscle injected.

## Figures and Tables

**Figure 1 toxins-14-00734-f001:**
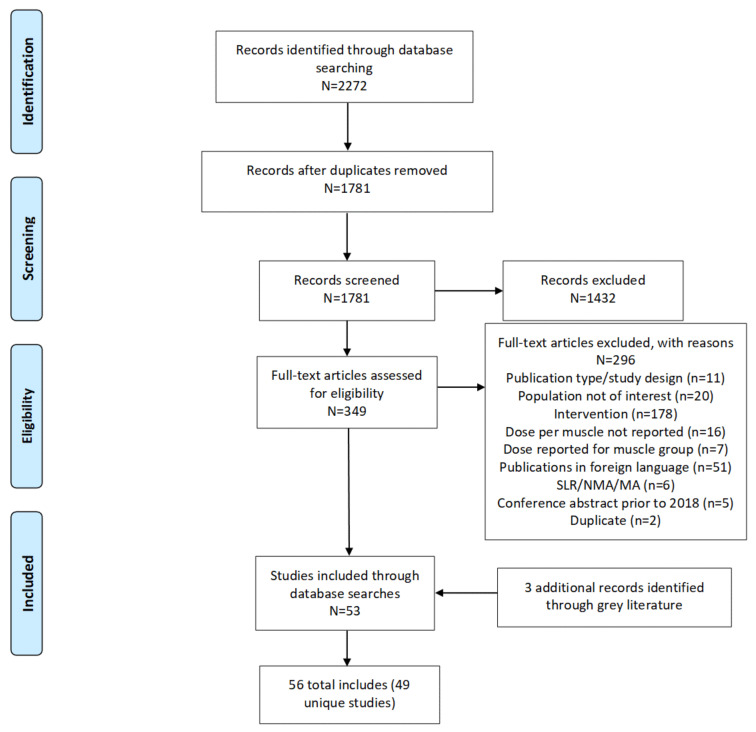
Preferred reporting items for systematic reviews and meta-analyses flow diagram. Legend: MA = meta-analysis; N = total number of records in the identified box; n = number of records in each category; NMA = network meta-analysis; SLR = systematic literature review.

**Figure 2 toxins-14-00734-f002:**
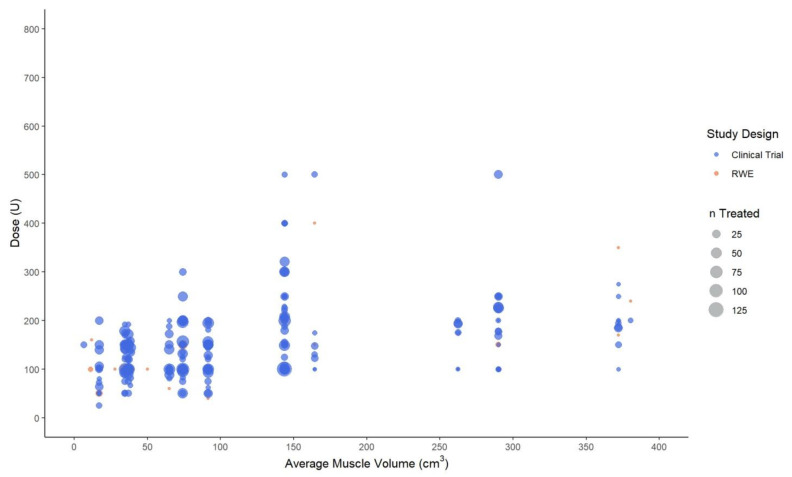
Mean, median and fixed abobotulinumtoxinA dose (in units) by the average volume of upper limb muscles. Legend: U, unit; n, number of patients injected with abobotulinumtoxinA in a specific muscle at a specific dose.

**Figure 3 toxins-14-00734-f003:**
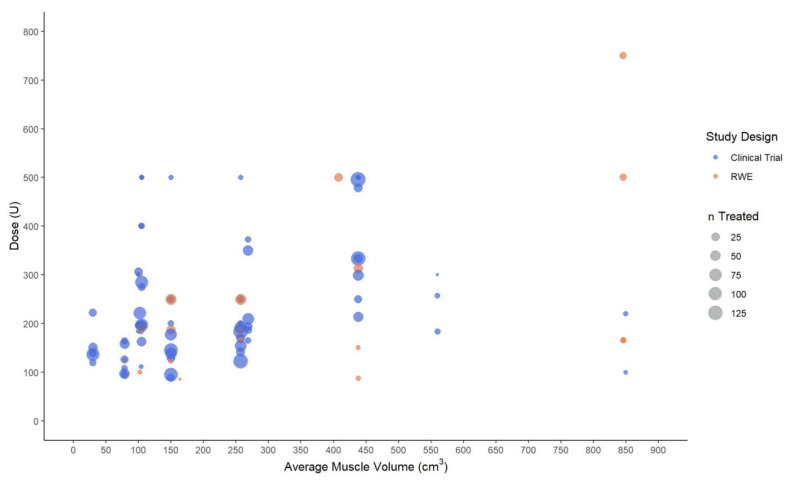
Mean, median and fixed abobotulinumtoxinA dose (in units) by the average volume of lower limb muscles. Legend: U, unit; n, number of patients injected with abobotulinumtoxinA in a specific muscle at a specific dose.

**Table 1 toxins-14-00734-t001:** Characteristics of the included studies.

Author, Year/Study Name	Country/Region	Study Design	Population Description	Sample Size/Enrollment Years ^1^
Alvisi, 2018 [10]	Italy	RWE	Subacute hemiparesis due to stroke	14/NR
Ashford, 2009 [11]	UK	RWE	Proximal ULS due to stroke or other acquired brain injury	16/2003–2006
Bakheit, 2000 [12]	International (Europe)	RCT	ULS due to stroke	83 (82 randomized)/NR
Bakheit, 2001 [13]	International (UK, Ireland, Germany)	RCT	ULS due to stroke	59/NR
Bakheit, 2002 [14]	UK	Single-arm trial	Attendees of an outpatient rehabilitation program with ambulatory hemiplegic stroke	9/NR
Bakheit, 2004 [15]	International (UK, Russia)	Single-arm trial	Established ULS due to stroke	51/NR
Barden, 2014 [16]	Australia	RWE	First onset of acquired brain injury with UL function affected by upper motor neuron syndrome	28/NR
Baricich, 2008 [17]	Italy	RCT	Chronic hemiplegia with spastic equinus foot	23/2005–2006
Beseler, 2012 [18]	Spain	RWE	Various brain or spinal cord injuries	10/NR
Bhakta, 1996 [19]	UK	Non-randomized trial	Severe spasticity and a non-functioning arm due to stroke	11/NR
Bhakta, 2000 [20]	UK	RCT	Stroke with spasticity in a functionally useless arm	54 (40 randomized)/NR
Burbaud, 1996 [21]	France	RCT	Hemiparesis with ankle plantar flexor and foot invertor spasticity	23/NR
Cardoso, 2007 [22]	Brazil	Single-arm trial	Spasticity with UL function disability due to stroke	20/2004–2006
Carvalho, 2018 [23]	Portugal	RWE	ULS due to stroke	86/2001–2016
de Niet, 2015 [24]	Netherlands	RWE	Hereditary spastic paraplegia with symptomatic calf muscle spasticity and preserved calf muscle strength	15 (+10 controls)/NR
Finsterer, 1997 [25]	Austria	RWE	Severe paraspasticity, limb spasticity or tetraspasticity	9/NR
Frasson, 2005 [26]	Italy	RWE	Spastic paraparesis following MS or other neurodegenerative conditions	12/NR
Ghroubi, 2020 [27]	Tunisia	RWE	Hemiparesis due to stroke or TBI	45/2014–2016
Gracies, 2017 [28]	International (Australia, Belgium, Czech Republic, France, Hungary, Italy, Poland, Portugal, Russia, Slovakia, USA)	RCT + OLE	Chronic hemiparesis due to stroke/brain injury with LLS	388/2011–2014
Gracies, 2018 [29]/ENGAGE	International (France, Czech Republic, Russia, USA)	Single-arm trial	Acquired brain injury	157/data cut-off December 2017
Gul, 2016 [30]	International	RCT (post-hoc analysis)	Hemiparesis	253/NR
Hecht, 2008 [31]	Germany	RWE	Hereditary spastic paraplegia	19/NR
Hesse, 1995 [32]	Germany	Single-arm trial	Hemiparesis with LLS due to stroke	10/NR
Hesse, 1998 [33]	Germany	RCT	Stroke	24/NR
Hubble, 2013 [34]	International (France, Germany, Greece, Sweden, UK)	RWE (survey of physicians)	Survey of physicians treating patients with ULS or LLS	275 physicians/July–September 2009
Johnson, 2002 [35]	UK	RCT	Stroke	32 (21 randomized)/NR
Kong, 2007 [36]	Singapore	RCT	Stroke	82 (17 randomized)/2002–2004
Lam, 2012 [37]	Hong Kong, China	RCT	Significant ULS and difficulty in basic UL care due to stroke or brain injury	55/January 2010–July 2010
Lejeune, 2020 [38]/AUL (open-label extension)	International (7 countries across Europe and in the USA)	RCT (OLE)	Stroke and TBI	254/NR
Marco, 2007 [39]	Spain	RCT	Stroke	31/August 2001–July 2003
McCrory, 2009 [40]	Australia	RCT	ULS due to stroke	102 (96 randomized)/2004–2006
Moccia, 2020 [41]	Italy	RWE	MS	386/September 2017–September 2018
Nott, 2014 [42]	Australia	RWE	Acquired brain impairment	28/NR
O’Dell, 2018 [43]/AUL	International (Belgium, Czech Republic, France, Hungary, Italy, Poland, Russian Federation, Slovakia, USA)	RCT	ULS > 6 months after stroke or TBI	243/2011–2013
Otom, 2014 [44]	Jordan	RWE	Stroke	26/January 2009–December 2009
Pauri, 2000 [45]	Italy	RWE	LLS due to MS or other neurodegenerative conditions	15/NR
Picelli, 2012 [46]	Italy	RWE	Patients with spastic equinus foot due to stroke scheduled to receive an AboBoNT-A injection into the gastrocnemius muscle	56/2010–2011
Picelli, 2014 [47]	Italy	RCT	Chronic stroke with wrist and fingers spasticity due to stroke	127 (60 randomized)/2011–2012
Picelli, 2016 [48]	Italy	RCT	Outpatients with spastic equinus due to chronic stroke	49 (22 randomized)/NR
Picelli, 2020 [49]	Italy	RWE	Patients with chronic stroke with spastic equinovarus foot attending a clinical neurorehabilitation unit	34/2016–2019
Rekand, 2019 [50]	International (Denmark, Finland, Norway, Sweden)	RCT	ULS due to stroke or TBI	88/2012–2015
Rosales, 2012 [51]/ ABCDE-S	International (Hong Kong, Malaysia, the Philippines, Singapore, Thailand)	RCT	Patients recruited within 2–12 weeks of first-ever stroke and upper extremity spasticity	163/2003–2007
Shaw, 2010 [52]/BoTULS	UK	RCT	ULS due to stroke	333/2005–2008
Sun, 2010 [53]	Taiwan	RCT	Chronic stroke with upper extremity spasticity	32/February 2005–November 2007
Suputtitada, 2005 [54]	Thailand	RCT	ULS due to stroke	50/NR
Turner-Stokes, 2013 [55]/ULIS-II	International (22 countries/Europe, Asia, South America)	RWE	ULS due to stroke	456/2010–2011
Woldag, 2003 [56]	Germany	Single-arm trial	Hemiplegia due to ischemic or hemorrhagic stroke	10/NR
Yazdchi, 2013 [57]	Iran	RCT	Stroke (ischemic or hemorrhagic documented by CT or MRI)	68/July 2010–December 2012
Yelnik, 2007 [58]	France	RCT	Hemiplegia with ULS due to cerebral stroke	20/NR

^1^ Number of patients enrolled in each study; this may also include patients receiving treatment other than AboBoNT-A. Legend: ABCDE-S, Asian Botulinum Toxin-A Clinical Trial Designed for Early Post-Stroke Spasticity; AboBoNT-A, abobotulinumtoxinA; AUL, adult upper limb; BoTULS, Botulinum Toxin for the Upper Limb after Stroke; CT, computed tomography; LLS, lower limb spasticity; MRI, magnetic resonance imaging; MS, multiple sclerosis; NR, not reported; OLE, open-label extension; RCT, randomized controlled trial; RWE, real-world evidence; TBI, traumatic brain injury; UK, United Kingdom, UL, upper limb; ULIS-II, Upper Limb International Spasticity Study-II; ULS, upper limb spasticity; USA, United States.

**Table 2 toxins-14-00734-t002:** Range of mean and median doses by muscle-volume categories across the included studies.

Muscle-Volume Category	Range of Muscle Volume (cm^3^)	Range of Dose Means (U)	Range of Dose Medians (U)
Upper Limb			
Small (<20 cm^3^)	6.6–17.1	47.0–150.0	25.0–200.0
Medium (20–99 cm^3^)	28.0–91.6	62.5–200.0	50.0–300.0
Large (≥100 cm^3^)	118.6–380.5	50.0–400.0	75.0–300.0
Lower Limb			
Small (<100 cm^3^)	30.0–78.8	94.9–233.3	NR
Medium (100–399 cm^3^)	100.1–269.0	85.0–372.7	NR
Large (≥400 cm^3^)	407.4–1803.0	88.0–495.3	NR

Legend: NR, not reported; U, unit.

## Data Availability

The datasets used and analyzed during the current study are available from the corresponding author on reasonable request.

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
