# Peer review of "AbobotulinumtoxinA Doses in Upper and Lower Limb Spasticity: A Systematic Literature Review"

_toxins, 2022, doi:10.3390/toxins14110734_

Round 1

Reviewer 1 Report

The manuscript is a systematic review about abobotulinum toxin A dosing for limb spasticity, focusing on muscle volume as the main criterion to standardise the amount of toxin to be administered. The methodology seems accurate, however, there are major issues to be considered.

- Abstract, "Disabling limb spasticity can result from stroke, traumatic brain injury or progressive neurological disease": the use of "progressive neurological disease" seems inappropriate here since there are many progressive neurological diseases that do not cause spasticity. Probably it should be replaced with something like "other disorders causing upper motor neuron lesions". Following suit, the second sentence of the introduction should be rearranged.

- line 113 "multiple sclerosis or other neurodegenerative conditions", as for the above comment, the Authors seem to mix up disorders affecting upper motor neurons with those characterised by a progressive course and also those with an underlying neurodegenerative mechanism (or inflammatory?). This shall be clarified.

- Table 1, first column: why the Study name (when available) is cited before the manuscript citation?

- The reported analysis about the amount of aboBoNT-A used in different spastic muscles is solely based on estimated muscle volume while ignoring the level of hypertonia. Of course, the amount of BoNT-A depends on muscle volume, but even more on how much that muscle is hypertonic. This must be taken into account when "administration of inadequate doses" is being discussed. Indeed, the Authors try to explain an obvious "variability of aboBoNT-A doses per muscle" without mentioning the amount of spasticity (lines 261-264).

- Recent works demonstrate how spastic dystonia is probably the most important form of muscle hypertonia due to upper motor lesions determining disability and therefore deserving BoNT-A treatment. Even if the study focuses on spasticity, the importance of assessing spastic dystonia (which is different from the mentioned "dystonia") should be mentioned. Furthermore, a more precise definition of spasticity based on the most accurate criteria (see Lance's definition of spasticity as velocity-dependent muscle hypertonia due to upper motor neuron lesion) should be provided.

Author Response

1) Abstract, "Disabling limb spasticity can result from stroke, traumatic brain injury or progressive neurological disease": the use of "progressive neurological disease" seems inappropriate here since there are many progressive neurological diseases that do not cause spasticity. Probably it should be replaced with something like "other disorders causing upper motor neuron lesions". Following suit, the second sentence of the introduction should be rearranged.

Response to 1: This point has been addressed by replacing "progressive neurological disease" with "other disorders causing upper motor neuron lesions, such as multiple sclerosis".

2) line 113 "multiple sclerosis or other neurodegenerative conditions", as for the above comment, the Authors seem to mix up disorders affecting upper motor neurons with those characterised by a progressive course and also those with an underlying neurodegenerative mechanism (or inflammatory?). This shall be clarified.

Response to 2:  “other neurodegenerative conditions” in the original sentence has now been replaced by "or other disorders causing upper motor neuron lesions (e.g., degenerative myelopathy, Strümpell-Lorrain disease)". 

3) Table 1, first column: why the Study name (when available) is cited before the manuscript citation?

Response to 3: As far as we can see, the study name (in bold) comes after the manuscript citation.

4) The reported analysis about the amount of aboBoNT-A used in different spastic muscles is solely based on estimated muscle volume while ignoring the level of hypertonia. Of course, the amount of BoNT-A depends on muscle volume, but even more on how much that muscle is hypertonic. This must be taken into account when "administration of inadequate doses" is being discussed. Indeed, the Authors try to explain an obvious "variability of aboBoNT-A doses per muscle" without mentioning the amount of spasticity (lines 261-264).

Response to 4: ‘Severity of hypertonia’ has now been added as a dose-variability factor in the Discussion and Clinical opinion sections of the manuscript.

5) Recent works demonstrate how spastic dystonia is probably the most important form of muscle hypertonia due to upper motor lesions determining disability and therefore deserving BoNT-A treatment. Even if the study focuses on spasticity, the importance of assessing spastic dystonia (which is different from the mentioned "dystonia") should be mentioned. Furthermore, a more precise definition of spasticity based on the most accurate criteria (see Lance's definition of spasticity as velocity-dependent muscle hypertonia due to upper motor neuron lesion) should be provided.

Response to 5: This very important point targets a common and potentially confusing misuse of the word “spasticity” in most clinical studies and even in the SmPCs of all toxins. In this manuscript, spasticity is used as a standard term to refer to the three components of muscle hypertonia: spasticity, spastic dystonia and spastic co-contractions. Therefore the definition of spasticity by Lance has been omitted.

Reviewer 2 Report

This a well written review article and a summary of the literature with reference to muscles treated with aboBoNT-A for spasticity and the dose used in muscles based on size/volume.  This is one of the factors that could be impacting the degree of response.  

Please consider addressing the following:

1.  Page 2, line 55:  Cerebral palsy listed as a progressive condition - I suspect that this was in error.  Please consider rewording.

2.  In the Introduction section mention other factors that could influence the dose of botulinum toxin used per muscle, such as severity of the spasticity, time post onset of spasticity, distribution of muscles affected, etc.  I know that this is mentioned briefly in the discussion, but consider expounding on this in there Discussion Section as well.

3. If possible, it would be good to see any information available from the studies with reference to dose and severity of spasticity/tone/MAS.  I realize that this may be difficult to obtain or make sense of.

4. In the Clinical Opinion section, consider adding severity of spasticity in the list of factors

Author Response

1. Page 2, line 55:  Cerebral palsy listed as a progressive condition - I suspect that this was in error.  Please consider rewording.

Response to 1: The sentence has been reworded to address this point.

2. In the Introduction section mention other factors that could influence the dose of botulinum toxin used per muscle, such as severity of the spasticity, time post onset of spasticity, distribution of muscles affected, etc.  I know that this is mentioned briefly in the discussion, but consider expounding on this in there Discussion Section as well.

Response to 2: Examples of factors influencing BoNT-A dose have been added in the Introduction.

3. If possible, it would be good to see any information available from the studies with reference to dose and severity of spasticity/tone/MAS.  I realize that this may be difficult to obtain or make sense of.

Response to 3: We have tried to extract and correlate other data such as severity of spasticity or safety with dose/muscle. However, these data were not available in all publications and it was complicated to correlate them with one muscle treated or the other.

4. In the Clinical Opinion section, consider adding severity of spasticity in the list of factors

Response to 4: Severity of hypertonia has been added in the list of factors.

Reviewer 3 Report

The authors performed a systematic literature review on the dosages of aboBoNT-A in the treatment of upper and lower limb spasticity. The objective of their work (as stated in the manuscript) was to gather evidence on intramuscular dosages of aboBoNT-A used by healthcare professionals. In table 3, they proposed some dosage ranges per muscle volume. Still, the caption specified that their proposals intended to trigger debate and facilitate consensus, not represent a clinical recommendation. Although the authors adequately designed and carried out the systematic review, they don’t provide how they intended to trigger the debate among healthcare professionals to obtain an expert-based consensus. 

In my opinion, the current version of the manuscript is an appropriate starting point for an in-depth analysis of the current treatment of upper and lower limbs spasticity, but it is not of interest to healthcare professionals. The authors should delete the statement “the proposals are intended to trigger debate and facilitate consensus, not to be taken directly as clinical recommendations” from table 3 and should improve the text by explaining in-depth why they propose these dosages, also trying to provide a risk-benefit assessment to justify the appropriateness of suggested dosage.

Author Response

1) The authors should delete the statement “the proposals are intended to trigger debate and facilitate consensus, not to be taken directly as clinical recommendations”

Response to 1: The text has been modified to more clearly reflect the intention of the proposal. Specifically, it now reads as follows: : "With this in mind and based on our practice, we propose “easy to remember “ narrow  aboBoNT-A dose ranges to be injected into muscles of different volume categories, as listed in Table 3. [...] These dose ranges are starting-points and the dose to be used may be adjusted based on the following factors: [...]"

Additionally, the text in the legend of Table 3 now reads as follow: "These proposals are intended to facilitate first intention aboBoNT-A treatment in botulinum toxin-naïve patients, not to be taken directly as clinical recommendations."

2) ...also trying to provide a risk-benefit assessment to justify the appropriateness of suggested dosage.

Response to 2: The text has been modified to reflect the intention of the authors to propose efficient but also safe dose recommendations. Specifically the text now reads as follows:   :   

"In general, we observed that the suggested dose is 1 to 1.5 times the muscle volume (100 to 150 U for a muscle volume of 100 cm3) for both upper and lower limbs. These rather conservative doses ranges have a well-established safety profile since they are within SmPc dose ranges (for in-label muscles). However, doses can be adjusted according to the efficacy and the desired effect. Dose increases are possible in the absence of safety concerns  and if there is insufficient effect from a previous dose. These dose ranges are starting-points and the dose to be used may be adjusted based on the following factors:"

Round 2

Reviewer 1 Report

Following my comments, the Authors submitted a mildly revised manuscript that however seems sufficiently accurate for publication.

Reviewer 3 Report

Now the manuscript can be published.